# Ageing society and the challenge for social robotics: A systematic review of Socially Assistive Robotics for MCI patients

**Giusi Figliano**[1]*, **Federico Manzi**[1,2], **Andrea Luna Tacci**[1], **Antonella Marchetti**[1,2], **Davide Massaro**[1,2]

**1** Department of Psychology, Research Unit on Theory of Mind, Università Cattolica del Sacro Cuore, Milan, Italy, **2** Department of Psychology, Research Unit on Robopsychology in the Lifespan, Università Cattolica del Sacro Cuore, Milan, Italy

* giusi.figliano1@unicatt.it

**Data Availability Statement:** All relevant data are within the paper and its Supporting Information files.

## Abstract

The aging population in Western countries has led to a rise in predementia conditions like Mild Cognitive Impairment (MCI). Social Assistive Robotics (SAR) interventions, among novel technological tools, offer a promising interdisciplinary approach to mitigate cognitive and social symptoms' progression in this clinical group. This systematic review aims to identify existing clinical protocols employing social robots for treating cognitive and social cognition skills in individuals with MCI. The review protocol adheres to the Preferred Reporting Items for Systematic Reviews and Meta-Analysis (PRISMA) guidelines. From six databases, we retrieved and analyzed 193 articles, of which 19 met the inclusion criteria, featuring samples diagnosed with MCI and subjected to cognitive and/or social interventions through SAR. The review encompasses both qualitative and quantitative studies, with a focus on assessing bias risk. Articles were categorized into four primary areas: study participants' samples, types of robots and programming used, assessment of cognitive abilities, and the nature of interventions (i.e., cognitive and and social cognition skills). While the findings highlight the potential benefits of using SAR for MCI interventions in both cognitive and social cognition domains, the studies primarily emphasized robot acceptability rather than intervention outcomes. Methodological limitations such as clinical heterogeneity, absence of control groups, and non-standardized assessments restrict the generalizability of these findings. This review underscores the promising role of Social Assistive Robotics in MCI interventions, emphasizing the importance of social cognition skills interventions and advocating for increased collaboration between clinicians and robotic researchers to overcome current limitations and enhance future outcomes.

## 1. Introduction

The World Health Organization revealed that in the world around 50 million people suffers of dementia and there are nearly 10 million new cases every year [1]. Most of these countries

**Funding:** This research and its publication are supported by the research line (funds for research and publication) of the Università Cattolica del Sacro Cuore of Milan and by "PON REACT EU DM 1062/21 57-I-999-1: Artificial agents, humanoid robots and human-robot interactions" funding of the Università Cattolica del Sacro Cuore of Milan. DM is the recipient of these funds. The funders had no role in study design, data collection and analysis, decision to publish, or preparation of the manuscript.

**Competing interests:** The authors have declared that no competing interests exist.

introduced several legislative proposals to consider the social and political implications of an ageing society [2]. National healthcare systems are increasingly interested in identifying and intervening at an early stage in neurodegenerative pathological conditions, on the one hand to prevent a progression of clinical conditions, leading to a decrease in quality of life, and on the other hand to reduce the economic and social burden on the healthcare system. In this sense, Mild Cognitive Impairment (MCI) represents a particularly important clinical population for the above-mentioned purposes. MCI is a clinical condition in which individuals experience cognitive decline with minimal impairment in instrumental activities of daily life (e.g., shopping independently, paying bills, typing telephone numbers) [3]. However, this clinical condition has been widely recognized to increase the risk of conversion to Alzheimer's disease (AD) or other severe neurodegenerative conditions [3,4]. To mitigate the incidence of conversion to other neurodegenerative conditions, it is important that specialized intervention strategies and compensatory home adaptations (e.g., digital aids for remembering appointments and tracking medication) are activated to reduce cognitive decline and maintain personal autonomy.

From the age of 65 years, between 10% and 20% of seniors may experience a condition of MCI [5] consisting of cognitive impairments in specific domains including memory, language, attention and visuo-spatial abilities [6,7]. Although not directly included in the diagnostic criteria, Theory of Mind (ToM) may also be affected by the general deterioration of the MCI symptomatology, leading to a considerable impairment of the subject's socialization skills [3]. ToM is the ability to understand own and others' mind and behaviors in terms of mental state (e.g., emotions, intentions, beliefs) [1,8], and it is essential for sociality and relationships. This ability is particularly sensitive to the cognitive and affective changes that occur in old age and may decline in its functioning with MCI [9,10]. The impairment of this ability negatively affects social life's activities and general well-being [2,6].

Considering the complexity of the clinical picture of MCI, an effective pharmacological treatment has not yet been identified. However, there are several non-pharmacological treatments that efficacy slow the progress of cognitive impairments preventing a further decline in different cognitive areas [11,12]. Specifically, cognitive training is particularly recommended with MCI patients [7–9,11–26]. Among the different cognitive trainings, Cognitive Stimulation (CS) is the one most frequently employed with MCI patients [12,26]. CS consists of cognitive exercise sessions to enhance residual cognitive abilities using the principles of neuronal plasticity [27]. Classically, cognitive stimulation exercises are administered paper-and-pencil by a clinician. As mentioned above, MCI also affects the socio-cognitive domain inducing an increase in behavioral symptoms [3] and negatively influencing the quality of life of the patients and caregivers [10]. A multi-stimulation intervention for patients with AD has been adapted for MCI, showing that ToM is an important measure for evaluating treatment progress on social cognition skills [10]. More generally, social impairments in MCI are treated through occupational/recreational activities secondary to interventions on cognitive domains [28]. These secondary interventions do not allow specific assessment on social cognition components.

In the last decade, cognitive and social interventions have been implemented through Social Assistive Robotics (SAR) to reduce and prevent increased MCI symptomatology [6,29–31]; to date, even though a significant increase in economic investment and the interest of the scientific and clinical community in these new types of interventions with SAR, it is not sufficiently clear their effects on cognitive decline and affective issues in this clinical population. In view of this lack of clarity regarding the benefits and efficacy of SARs in cognitive and social interventions with MCI patients, this systematic literature review aims to analyze clinical studies that employed SAR for cognitive and social interventions for MCI.

## 2. Methods

A systematic review of scientific literature has been performed to identify studies that reported research which used social robots for cognitive and social interventions for MCI. A review protocol was compiled, following the Preferred Reporting Items for Systematic Reviews and Meta-Analysis (PRISMA) guidelines [14].

### 2.1. Data sources and search strategy

Electronic literature searches were performed using ACM Library, Cochrane Systematic Reviews Database, Google Scholar, PubMed, PsychINFO and Web of Science including publications up from January 2000 to March 2022. Two researchers reviewed the potential studies individually for eligibility. Was used a list of keywords to identify the studies, including relevant interventions, through an interactive process of search and refine (see Table 1). Each database was searched independently, according to a specific interaction research string: ("Mild Cognitive Impairment" OR "MCI") AND ("Robot" OR "Robots" OR "Human robot interaction" OR "Humanoid robot" OR "Zoomorphic robot").

No limitations regarding study design or outcome measures were used. We included only English's article. Eligible studies were those whose title or abstract specifically indicated the inclusion of MCI, use of social robots and protocol of cognitive and social interventions. There were no restrictions about the age and number of participants. In the first export were included only article with full-text available. The complete list was exported in EndNote to remove duplicates and then it was imported in Rayyan [29] for title and abstract screening.

### 2.2. Study selection criteria

The aim of this review was to evaluate the efficacy of social robot-based interventions with MCI patients to enhance cognitive and social abilities. The following selection criteria were applied to the articles found in databases: research studies, reviews and case reports were eligible for inclusion; chapters and not peer review studies were excluded. The abstracts of the identified publications were screened for relevance to the selection criteria. Specific inclusion criteria were: samples with MCI clinical condition; a cognitive and/or social training through a social robots. Papers that described unstructured and holistic intervention programs for patients with MCI were rejected.

**Table 1. Detailed search strategy.**

| Mild Cognitive Impairment OR MCI AND | PubMed | Cochrane Systematic Rewiev database | PsychInfo | Web of Science | ACM Library | Google Scholar | |
|---|---|---|---|---|---|---|---|
| Robot | 46 | 20 | 0 | 79 | 83 | 17600 | |
| Human Robot Interaction | 8 | 10 | 0 | 19 | 43 | 1830 | |
| Humanoid robots | 1 | 1 | 0 | 9 | 14 | 21200 | |
| Zoomorphic robots | 0 | 0 | 0 | 0 | 0 | 624 | |
| SubTotal | 55 | 31 | 0 | 107 | 140 | 41254 | |
| Total | | | | | | | 41587 |
| Duplicated removal | | | | | | | 333 |
| Identified studies for Abstract and Title screen | | | | | | | 193 |
| Included | | | | | | | 19 |

## 2.3. Quality assessment

The 'Tool for assessing the risk of bias in randomized quantitative trials' [32], which includes five domains related to the quality of the methodology in randomized trials, was used to assess the quality of the risk of bias (RoB2). Each domain was rated by two independent reviewers (GF, AT) who rated each domain by assigning it a risk of bias rating; the rating could be categorized as follows: low risk of bias; some concerns and high risk of bias. A judgement of 'High' risk of bias for any individual domain will lead to the result being at 'High' risk of bias overall, and a judgement of 'Some concerns' for any individual domain will lead to the result being at 'Some concerns', or 'High' risk, overall. With the purpose of calculating the risk of bias for the qualitative studies included in this systematic review, it was decided to use the tool of GRACE (Good Research for Comparative Effectiveness) that is an 11-item instrument designed to evaluate the quality of the data and the methods used either in the design and the analysis of the noninterventional and observational studies of comparative effectiveness [33]. The GRACE tool is composed by two sections: the first dedicated to the data (6 items) of the study regarding the treatment, outcomes, and population; the second dedicated to the methods (5 items) with more information about the population, the control group, possible follow up etc. [33].

## 2.4. Quantitative studies

Concerning quantitative studies and the dimension related to the randomization process (S1 and S2 Figs), some studies [17,34–36] have a low risk of bias; this suggests a random distribution of participants to the different intervention groups and the difference in outcome between the two groups is not an indicator of randomization problems. Two studies [18,24] show some doubts, while works [16,23] have a high risk of error related to the randomization process.

The second aspect explored was the deviation from the intended interventions; here only study [35] shows a low risk of bias while four works [15,18,34,36] show some concerns regarding the adopted methodology. The remained works [12,16,24] on the other hand, present a high risk of bias; this proportion is related to the awareness of the subjects and their caregivers about the intervention and to factors related to the experimental context that may have affected the outcome of the intervention.

The third dimension of Rob2 concerns bias due to the lack of outcome data; this could happen when participants withdraw from the study 'dropout' and if they do not provide relevant data; if participants die, etc. In this case, the studies can be divided into low risk of bias: [35], (relating to the outcome of improvement in visual memory), [16–18,23,34,36] and high risk of bias: [35] related to the outcome of the improvement in executive functions, [35] in relation to increased cortical thicknesses, [18] in relation with change of prose memory, [12] concerning the increase in the frequency of communication in the robot group and the increased, in both groups, of interaction with staff, and [24].

Also, with regard to measurement of the outcome, one could categorize the work into studies with a low risk of bias: [12,17,24,34–36] and studies with a high risk of bias: [35], relating to the improvement in the executive functions in robot group, [12,16,18]. When referring to measurement error, this concerns misclassification (for dichotomous or categorical outcomes), the use of non-adapted tools for the measurement of the outcome being studied, the possibility that administrators may be aware of the intervention provided to subjects, etc.

Regarding the selection bias of the reported result, a dichotomous situation was found in studies [17,35] and [12] show some concern on the reported results, while a high risk of bias was observed for the studies [35] relating to the improvement in the executive functions in robot group, [12,16,18,24,34,36] with respect to the increase in frequency of positive

expressions and reductions of loneliness feelings in the robot group. With regard to this dimension, several studies do not report pre-specified analysis plans that were finalized before unblinded outcome data were available for analysis and lack multiple eligible outcome measurements from which to derive data.

As a final result, a global calculation of the risk of bias relating to each job can be observed; in this case, some results lead to an uncertain risk of bias [17,34–36] and in overall results with a high risk of bias [35], regarding improvement in visual memory, [12,16,18,24].

## 2.5. Qualitative studies

With regard to data on qualitative studies, two independent assessors compiled the GRACE scale (S3 Fig) [33] and then compared the results and reached an inter-judge agreement. The Grace scale is subdivided into two macro sections: a first section on data and a second section on methods [33]. We will begin the analysis of the results of the papers included in the review from the first section which is concerned with investigating whether the study data are adequately recorded (D1-D2), whether they are clinical outcome (D3), whether they have been validated on similar populations in terms of diagnosis (D4), and whether the data have been measured on a comparison group (D5). What emerged from the evaluations was that all studies correctly recorded data except the one of [37] where, although both qualitative and quantitative measures were implemented in the study, the latter were recorded through methods that are closer to a qualitative and observational methodology. Moreover, the contents of qualitative survey measures such as comments and open-ended questions are not specified in the paper, therefore, there is insufficient information in the publication to allow us to say whether the treatment data has been adequately recorded. Moreover, most of the studies did not have a clinical outcome expect for the following works: [5,22,38]. Most of studies were pilot studies or focused on new types of populations, the only studies that have used a protocol or previously validated measures in other populations are [39–41]. Regarding the reproducibility of the results in a hypothetical control group, we find that the only one studies have this condition [38]. Finally, the recording of important covariates was not found in any of the included work. Overall, what emerges was that, limited to the data section, most of the studies mentioned presents what could be defined as a high risk of bias expect for one work [38], which could be interpreted as a study with a medium risk of bias. Considering the methods section, we found how for the first item (M1) all studies, except for [38], included new initiators instead a population already under treatment, whereas, for the second item (M2) the studies which used an historical comparator group were: [20,38,40–42]. For the third item (M3) the studies which take into account important confounding and effect-modifying variables were: [40,41].

Finally, none of the studies was free of "immortal time bias" (M4) and none of them has conducted meaningful analyses to test the key assumption which primary results were based (M5). In conclusion, regarding methods section, all the studies included presented what we could define as a high risk of bias.

These results suggest how the novelty of the used protocols in 'human-robot interaction' studies, particularly in the rehabilitation context, are still in the development and validation phase all these aspects would be discussed and deepened later.

## 3. Results

After removal of duplicates, title and abstract screening of electronic database search results and identification of eligible articles through other sources, 193 articles were full text screened. A total of 19 articles were included and 4 systematic reviews was consulted (see Table 2 for the summary of the studies). See Fig 1 for study selection flow chart. In this review were included:

**Table 2. Papers about evaluation.**

| Authors | Research goal | Subjects | Type intervention | Assessment | Results | Limitations |
|---|---|---|---|---|---|---|
| [35] | Compare traditional cognitive training with robot-assistive cognitive training | 48 participants from 60 years old with starting symptoms of cognitive impairment | 24 participants: traditional cognitive training 24 participants: Robot (SILBOT and MERO) cognitive training Training: multi-domain exercises | Pre and post neuropsychological assessment and MRI: Alzheimer's Disease Assessment Scale (ADAS-Cog); Cambridge Neuropsychological Test Automated Battery (CANTAB) | Greater improvement of visual memory in traditional group Improvement in the robot groups in executive function performance | It was impossible to control participants' daily cognitive activity at home The presence of more females than males within the sample |
| [39] | Improve the quality of life of elderly individuals with moderate dementia and/or depression through conversation and cognitive games | 6 participants with moderate dementia and/or depression in a Senior Community | Memory games mediated by a robot, RYAN, with personalized contents (like quiz, music and video); they could give oral answer or through tablet | Observation and interviews to caregivers and participants; interaction's analysis; they used two geriatric scale for subjects: SLUMS e PHQ9 | It was a good acceptance of robot by elderly people and an improvement of participants' mood (reported by caregivers) | Exercises were too easy for subjects with initial cognitive impairment |
| [38] | Memory games to train memory's functions through a humanoid robot, PEPPER, and tablet. Authors want to understand subjects' preference | 14 participants, age above 65 years old and diagnosis of MCI | Musical quizzes: subjects had to recognize sings or singers. One group train with PEPPER and one with Tablet | Likert scale to understand satisfaction's grade of participants and caregivers | All participants ended without difficult musical quizzes; subjects and caregivers were satisfied | There aren't multiple difficulty levels |
| [16] | Explore the robot's potential to engage participants in the intervention and its effects on their emotional state | 21 patients with MCI, aged between 45–85 years old | Therapist assisted by NAO propose memory training to little group | Neuropsychological assessment (a complete test battery) ADL and IADL | In the subjects are generated positive emotions towards robot and they consider it as if it was a real companion with real intentions | Small sample and short time for the study |
| [18] | Evaluate the effectiveness of human–robot interaction to reinforce therapeutic behavior and treatments adherence and improve memory functions | 21 subjects MCI; 45–85 years old | There are two groups: one group did memory training mediated by NAO and one group did traditional training with psychologist | Neuropsychological pre and post intervention: digit span test, prose memory and fluency; clinical assessment: anxiety and depression | Training with NAO resulted in an increase of visual gaze from patients and reinforce of therapeutic behavior depressive symptoms. Changes in prose memory and verbal fluency | Results not generalizable due to small sample size |
| [36] | Demonstrate the effects of our newly developed home-based cognitive intervention with robot BOMY on cognitive function in MCI patients | 46 patients with MCI; there are two groups: robot group and control group | 5 programs for home-based multi domain cognitive training for four weeks | Seoul Neuropsychological Screening Battery | Improvement of working memory in robot group | Larger samples and longer study periods are required to demonstrate the effects of these programs |
| [17] | It investigated whether multi-domain cognitive training programs, especially robot-assisted training, could improve cognitive function and depression decline in community-dwelling older adults with MCI | 135 volunteers with cognitive impairment aged 60 years old or older There are two group: one robot-group and a control group that do traditional cognitive training | Multi-domain cognitive training conducted by SIL-BOT for 12 times, twice a week for 6 weeks | MMSE-Ds, Cerad-K, Sgds-K | Robot-assisted cognitive training group had significantly greater post-intervention improvement in memory, executive functions and depression. Traditional cognitive training group had improvement in memory and executive functions | Lack of integrated approach for improving the physical and emotional functions of the elderly; gender, age, and years of education affect the effectiveness of training program |

*(Continued)*

**Table 2.** (Continued)

| Authors | Research goal | Subjects | Type intervention | Assessment | Results | Limitations |
|---|---|---|---|---|---|---|
| [15] | To preliminarily evaluate how acceptable robot-mediated pet-therapy is for older people with light cognitive deficits (MCI) | 24 subjects with aMCI and more advanced degrees of decay | Experimental sessions in which the patient interacts with AIBO (through operator mediation) and answers questions related to the potential use of the robot | Mini Mental State Examination | AIBO is perceived as friendly | Preliminary study |
| [20] | Evaluation the seal-like robot PARO in the context of multi-sensory behavioral therapy (MSBT) | 10 elderly nursing home residents with varying levels of dementia (from mild to moderate) | PARO engaged participants through multimodal sensory stimulus in group therapy, one a week for seven weeks | Researchers observe and videotape interactions | Increase of verbal communication with PARO and of the interaction between participants | Small sample size and short period of interaction |
| [21] | Observe acceptance of a zoomorphic robot, PARO, as a companion to reduce sense of loneliness | 30 subjects in single room in an elderly's residence, someone with mild/moderate dementia (19), some with severe dementia (11) | 18 months, every 3–6 months individual sessions for 15 minutes; an experimental group interacts with PARO and a control group interacts with a stuffed animal (Lion) | Hasegawa's Dementia Scale to assess the level of dementia. Video recording interactions and recording the frequency of certain behaviors | In the robot group, an increase in the frequency of positive expressions is observed and a reduction in the feeling of loneliness is reported. Subjects talk more to PARO than to Lion; in both groups, is observed more interaction with staff | Small sample |
| [24] | Compare the effect of different rehabilitation sessions with different modalities and different robots: NAO and PARO | Group sessions (9–15 persons) for mild to moderate dementia, individual sessions for more severe dementia | 3 times a week for 3 months. Various types of activities: sensory, cognitive, socialized, and different levels of difficulty | Neuropsychological assessment pre and post training with MMSE, sMMSE and NPI | Robot-managed daily routine could support and reassure individuals | Several participants left the center or unit or died, and several patients joined the study late |
| [40] | The aims to show how the engagement between two social robots, SOPHIE and JACK, in Australian residential care facilities can improve care quality | 139 participants, 65–90 years old, (43 males, 96 females) with different stages of cognitive impairment in an elderly care facility | Designed to communicate in speech mode, touch panel | Behavioral reactions observed: approaching the robot in a positive way; the pleasure during interaction with the robots; interaction frequency with robots and interaction frequency with other staff and/or residents | These innovative social robots could improve the quality of care for people suffering from dementia | There isn't an objective neuropsychological and ToM assessment |
| [41] | Focuses on the service design and the effectiveness of the engagement and acceptability while interacting with a social robot | 115 participants in Australian residential aged care: with dementia aged 65–90 years; the participants had mild to advanced dementia | MATILDA has been designed to communicate in speech mode, touch panel mode, and facial recognition mode; it proposed games, musical quizzes, orientation activities | The measures of engagement were coded based on the guideline for video coding of engagement proposed by Jones et al. (2015). Emotional engagement in people with dementia was assessed via facial emotional responses based upon a modified version of the observed emotional rating scales of Lawton | An increase of involvement respect to the baseline | There isn't an objective neuropsychological and ToM assessment |

(*Continued*)

**Table 2.** (Continued)

| Authors | Research goal | Subjects | Type intervention | Assessment | Results | Limitations |
|---------|---------------|----------|-------------------|------------|---------|-------------|
| [22] | Evaluates effect on behavioral and psychological symptoms of KABOCHAN in subjects with MCI | Subjects: 74 elderly people aged ≥ 65 years with mild impairment, residents in elderly care facilities | Living with a communication robot | Administration of: IADL, STAI, QOL; administration of questionnaires before and after two months following interactions | Sleep, nutrition and conversation improve after one month of living with the robot. Reduction in anxiety after one month. Physical functioning improves after one month | There isn't an objective neuropsychological and ToM assessment |
| [23] | Facilitating conversations between a social humanoid robot, NADINE, and cognitively impaired elderly at a nursing home. We analyzed the effectiveness of human–humanoid interactions between our robot and elderly, to promote their emotive, cognitive and social impairments | 14 elderly people with cognitive impairment in a nursing home—One-to-one interactions | NADINE could talk, recognize and answer to resident's emotions. Robot may personalize arguments' core thanks the memorize information of the subjects. Residents could ask to NADINE to listen music or watching video | Pre and post assessment: Deep Neural Networks (DNNs); Observed Emotion Rating Scale (OERS); Menorah Park Engagement Scale (MPES) | An improvement of residents' wellness and cognitive skills; increased productivity by augmenting or reducing human resources | There isn't an objective neuropsychological assessment |
| [37] | Social robot PEPPER provides the music which supports positive self-disclosures of personal memories | 7 individuals with dementia and their caregivers | Group's interaction with PEPPER, participants and their caregivers. Listening to music, changing songs and creating a personalized playlist. Stimulating the evocation of memories and their narration by increasing emotional involvement | There isn't a neuropsychological or ToM's assessment | Elicit positive responses and individuals with dementia understand everything that is said | The participants suffered from slurred speech and often PEPPER wasn't able to understand them properly; the explanations given by PEPPER were too long for them to stay focused; it was not always clear to the participants what actions they were asked to perform or not perform |
| [34] | Evaluation of the level of involvement of participants in the activities offered by the KABO-CHAN robot | 103 participants diagnosed with moderate/severe dementia residing in elderly care facilities: age 67–108 | Cognitive stimulation exercises in the form of quizzes | The cognitive level was assessed by Hong Kong Montreal Cognitive Assessment 5-minute Protocol (MoCA) | Specifically, resident-robot behavioral engagement moderately improved attitudes towards technology perceived usefulness | Clinical heterogeneity of the sample precludes generalization of the results |
| [25] | Exploring the use of a home-based robot JAMES, during lockdown from COVID-19 to evaluate its use in cognitive activities and loneliness reduction | 4 elderly people diagnosed with MCI living in semi-autonomous housing (age 70–90) for a duration of two weeks | Subject interaction activities (robot implemented following a preliminary interview on subjects' interests) | Pre-test: 5-point questionnaire; post-test: semi-structured interview | Reduces feelings of loneliness and social isolation and is a motivator and facilitator in cognitive activities | Small number of subjects and short trial duration |
| [42] | Robotic architecture system with NAO to engage pairs of older adults in multimodal activities to reduce apathy | Seven pairs (14 individuals); ages ranged from 70 to 90 years. Three adults were screened as having normal cognition, 10 had mild cognitive impairment, and 1 adult self-reported a diagnosis of Alzheimer's disease | Each activity had a physical, cognitive and social components, 3 weeks for 6 sessions | MOCA scores to classify the individual as possible mild cognitive impairment or dementia (<19). Cohen-Mansfield's Observational Measurement of Engagement | Engagement measures (visual, verbal, behavioral) varied by type of activity; SAR activities had positive impact on engagement | Possible presence of apathy was not examined. The pairs remained the same throughout the three weeks and familiarity with one another may have impacted the engagement level. One participant opted out due to a physical limitation |

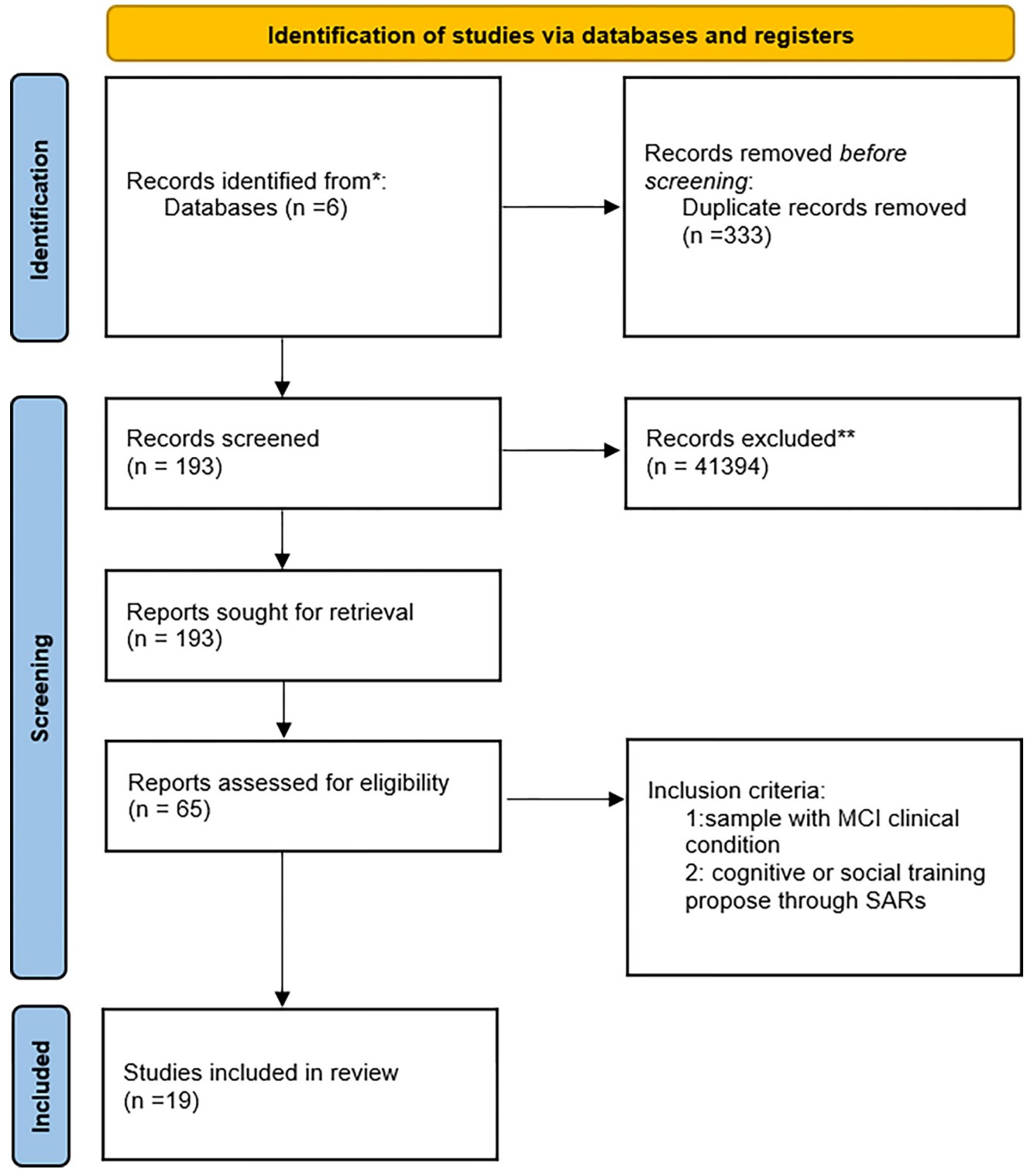

**Fig 1. Research methodology for review process.**

nine Randomized Control Trial; nine qualitative observational studies; one pilot study. The results are organized with respect to the following sections: characteristics of recruited samples; types of assessment for clinical evaluation and intervention outcomes; types of robots; and aim of the interventions, subdivided into cognitive and social.

### 3.1. Participants

A first important finding relates to the clinical pictures that compose the study samples. As a matter of fact, only seven studies involved people with a diagnosis of MCI [16,18,25,35,36,38], while the remaining twelve included mixed samples (i.e., MCI and patients with mild/moderate/severe dementia) [15,17,20,21,23,24,34,37,39–42].This result highlights that the selected studies do not have homogeneous samples of MCI, despite the studies declared as aim the exploration of the effects of interventions with social robots in this clinical population. Underlining the lack of clarity of the studies with respect to sample recruitment, only in one of the nineteen studies reported the specific cognitive area of impairment (i.e., anamnestic MCI; [18]).

Thirteen studies administered cognitive or social training to MCI without control groups [15,16,18,20,22–25,34,37,39–41] while six studies recruited control groups [17,21,35,36,38,42] that, however, they did not compared MCI patients involved in robot-based interventions with other MCI patients without robot-based interventions. This finding shows that the lack of control groups may reduce a real evaluation of the positive effects on maintaining and/or improving cognitive and social skills in MCI.

Another relevant aspect is where the interventions with MCIs were conducted. Eleven studies were carried out in residences for the seniors or nursing homes [17,20–24,34,37,40–42], two studies offered a home-based protocol [25,36], and in six studies the interventions were conducted in specialized centers for cognitive disorders and dementia [15–18,35,38]. This is relevant because the outcome of an intervention may vary depending on contextual conditions, again underlining the generalization problem of these studies.

Finally, the interventions were conducted in different countries: four in Italy [15,16,18,38]; three in Korea [17,35,36], two in Australia [37,41], Japan [21,22] and USA [20,39]; one in Belgium [25], China [34], Ireland [42], Netherland [37], Singapore [23] and Spain [24]. Cultural differences also could have an important influence on the clinical conditions and the practices to approach to them. This is an additional element that prove the heterogeneity of this filed of research.

### 3.2 Type of robots and programming

Another interesting issue that characterizes most of the studies is the heterogeneity of social robots used. Fifteen studies used humanoid robots (BOMY, JACK, JAMES, MATILDA, MERO, NADINE, NAO, PEPPER, RYAN COMPANIONBOT, SILBOT and SOPHIE) [16–18,23–25,35,35–42] and three used zoomorphic robots (AIBO and PARO) [15,20,21]. The heterogeneity of the social robots employed in the various studies does not allow generalization of the data to social robots and remains open if a specific social robot is more effective with MCI than others. One study used both the humanoid robot NAO and the zoomorphic robot PARO [24]. One study compared the effectiveness of two different types of humanoid robots (SILBOT and MERO) [35], and one study used two humanoid robots simultaneously (JACK and SOPHIE, [40]). Although these few studies used different social robots and claimed in the objectives to compare the effectiveness of these, the analyses did not report any specific results on this issue, thus leaving unanswered which robot is more effective. Regarding cognitive interventions, all the studies used anthropomorphic social robots: one the BOMY [36], one

both MERO and SILBOT [35], two the NAO [16,18], one the PEPPER [38], one the RYAN COMPANIONBOT [39] and one SILBOT [17]. This result highlights the importance of anthropomorphic features of robots for interventions that focus on the residual cognitive functions of MCI. However, there is no detailed consideration with respect to which anthropomorphic features are most functional for this clinical condition.

Regarding social interventions, eight studies proposed the use of anthropomorphic social robots: two the KABOCHAN [22,34], one the JAMES [25], one the MATILDA [41], one the NADINE [23], one the PEPPER [37], one the NAO [42], one both JACK and SOPHIE [40]. Four employed zoomorphic robots: one the AIBO [15], two with PARO [20,21], and one both the NAO and PARO [24]. The use of these two types of robots, anthropomorphic and zoomorphic, evidences a greater openness in interactions by MCIs toward robot design features in the emotional-relational sphere. This seems due to the caring behaviors that zoomorphic robots solicit in MCIs by supporting residual social skills.

All studies used the Wizard of Oz technique, and no autonomous interaction system was developed. This underlines how, to date, it is difficult to hypothesize interventions with social robots that can be actively applied by non-technicians in MCI care settings as well as within the homes of people with this clinical condition.

## 3.3. Assessment

Regarding the assessment of cognitive abilities of MCI, there is a wide variability of psychometric tests in literature and there are no standardized assessment protocols. In general, MCI neuropsychological assessment aims to identify the presence of specific cognitive impairments, quantify the severity of the disorder and, with respect to interventions, verify their effectiveness [19]. Regarding Theory of Mind abilities in MCI, they are measured through different classical tasks and tests (e.g., False Belief Tasks, Strange Stories, Reading the Mind in the Eyes) [13,43].

Regarding cognitive interventions, of the seven selected studies, only two studies used pre- and post-intervention neuropsychological assessment. One study used Cambridge Neuropsychological Test Automated Battery (CANTAB) [35], while the other one assessed episodic memory, short term memory and verbal fluency [18]. Of these two, only one assessed patients pre- and post-intervention with MRI [35]. All of the remaining five studies did not adopt a post-intervention assessment. Two studies specified the neuropsychological batteries used: specifically, the Seoul Neuropsychological Screening Battery [36], and the Mini-Mental State Examination—Dementia Screening (MMSE-DS) and Consortium to Establish a Registry for Alzheimer's Disease Korean version [17]. One study adopted a pre-intervention neuropsychological complete battery and two functional scales (Activities of daily living and Instrumental activities of daily living [16]). Other two studies used clinical observations of the participants and clinical interviews with participants and caregivers. Of these two, one specified the observational protocol and clinical interviews (i.e., Saint Louis University Mental Status Examination to detect mild cognitive impairment; [39]) and the other one stated that they analyzed neurocognitive characteristics without specifying the tests, trials, and/or interviews adopted [38].

With respect to social interventions, of the twelve studies includes, only three studies conducted a pre and post-intervention neuropsychological assessment: using the Mini Mental State Examination (MMSE), Severe Mini Mental State Examination (sMMSE and Neuropsychiatric Inventory, NPI) [24]; emotional through analysis of interactions using Deep Neural Networks (DNN) techniques and Menorah Park Engagement Scale (MPES) [23]; individual sense of well-being International Quality of life Assessment (QOL SF-8) and residual skill level Tokyo Metropolitan Institute of Gerontology Index of Competence (TMIG-IC) [22]. Other three studies did only an initial screening of cognitive competence through the use of MMSE

[15] and Montreal Cognitive Assessment (MoCa [34,42]). The remaining six studies did not explicitly state the assessment procedures because the objective primarily was to study the acceptance of social robots by MCI patients. Of these six studies, three conducted only an observational assessment [20,40,41], one assessed the level of dementia through a scale [18], one proposed a pre and post intervention questionnaire and semi-structure interview [25] and, finally, one didn't declare a neuropsychological or ToM's assessment [37].

Overall, quantitative assessment protocols of both cognitive and social interventions are extremely heterogeneous, lack standardized procedures, and, in addition, only a few studies present pre- and post-intervention evaluation. From a clinical perspective, these limitations prevent a full understanding of the actual benefits of the interventions (e.g., amnestic, language). At the same time, qualitative evaluations present similar issues since neither standardized procedure are applied nor observation protocols stated.

## 3.4. Type of training

**3.4.1 Cognitive interventions.**   Cognitive interventions are non-pharmacological treatments that employ specific cognitive exercises with the aim of slowing down cognitive decline and enhancing residual abilities. Of the seven studies that declared to conduct cognitive interventions, the main objective of five studies was to evaluate the effectiveness of using social robots on maintaining and/or enhancing cognitive function [17,18,35,36,38] while two studies aimed to evaluate the acceptance of social robots with MCI [16,39]. No study has thoroughly detailed the intervention protocol implemented in the social robots.

One study on acceptance proposed memory games with personalized contents (e.g., quiz, music and video) through RYAN COMPANIONBOT [39]. Although the robot was used for long-term memory exercises, the main goal was to investigate the acceptance of the robot. Regarding the cognitive training no significant results were reported, while for acceptance the results showed that users established a good relationship with the robot (i.e., accepting it as a companion) and it positively affected MCI's general mood. The other study on acceptance explored the potential of the robot to engage MCI during cognitive intervention and an evaluation of the effects on acceptability and emotional involvement in the interactions with the robot [16]. The training consisted of memory tasks (i.e., story reading, story comprehension questions, word learning, word recall, and song-singer matching) in which the NAO asked participants to respond verbally to its questions. The NAO was equipped with a system designed to analyze participants' facial expressions during the training. The results showed an increase in positive emotions toward the robot and involved it within their activities as if it were an interactive partner. Another study compared the acceptability of memory training (i.e., music quizzes in which patients have to recognize singers or vocalists) by comparing PEPPER and a tablet [38]. In both cases, users enthusiastically completed the training sessions, but no differences were found between the two modalities, robot and tablet. Although these studies claimed to target the efficacy of cognitive interventions, the protocols were constructed to primarily assess the acceptability of social robots by patients with MCI by overshadowing cognitive assessments.

With respect to the studies that directly trained cognitive functions, one study compared a multi-domain cognitive training administered by a clinician and by two different social robots, SILBOT and MERO [35]. Three groups were compared: a first one that did the training with the clinician, a second one with the social robots, and a control group. The results showed compared with the control group an improvement in executive functions alone in the social robot group, while an improvement in general cognitive abilities in the group with the clinician. Although these results are interesting, the study did not report possible effects due to the

two different robots. Another study proposed memory training by comparing the intervention of a clinician and the NAO [18]. In addition, treatment adherence in the robotic condition was evaluated. The results found both an improvement in various cognitive abilities (i.e., memory, attention, and verbal fluencies) and a decrease in depressive symptoms in the NAO condition. In addition, participants adhered positively to NAO treatment. Another study compared the effectiveness of multi-domain cognitive training when performed by a clinician or the SIL-BOT robot [17] and its effects on depressive symptoms of MCI. The results found a significant improvement in some cognitive domains (i.e., memory and executive functions) in both the condition with the clinician and the robot, while an improvement in depressive symptoms only in the condition with the robot. Finally, another study evaluated the effects of a home-based multi-domain cognitive intervention (i.e., memory, language, computation) developed with the BOMY robot [36]. The results showed an improvement in working memory in the group with the robot compared with a control group without cognitive intervention. Cognitive interventions that have pursued the goal of analyzing their effects on different cognitive domains revealed that robots positively affect at least memory and executive functions similarly to human clinicians. In protocols where different robots have been used, there are no data on possible effects due to the type of robot. Finally, improvements in mood are also noted as secondary effects of the interventions.

**3.4.2 Social interventions.** Interventions aimed at improving social abilities through SARs have been implemented either through anthropomorphic robots or zoomorphic robots in sessions of interactions between single patients and social robots or between groups of patients and social robots. Studies will be presented below according to these two main variables: type of robot (anthropomorphic and zoomorphic) and individual or group sessions.

Regarding the studies that used zoomorphic robots, one study evaluated the acceptability of pet-therapy mediated by the AIBO robot [15]. In individual sessions, patients interacted with the AIBO and were asked to give their opinion about their interaction with it. Analysis of the interviews revealed a friendly perception of the AIBO by the MCI. Another study compared the reduction in feelings of loneliness in one-to-one sessions at a specialized care center for the elders [21] by comparing PARO with a stuffed animal (i.e., Lion). Participants were free to interact in their own room with either PARO or Lion. Qualitative analysis of the interactions showed a greater increase in positive emotional expressions and verbal interactions and a greater reduction in feelings of loneliness in the group with the PARO than in the group with the Lion. Another study with PARO developed a multisensory therapy (physical, visual and verbal) with patients with different levels of dementia including MCI within residences for the elders [20]. Participants interacted in small groups in which PARO was present and the therapist mediated interactions between patients and between patients and PARO. Results showed an increase in verbal communication with PARO and interaction between participants. The use of zoomorphic robots was employed with heterogeneous samples including MCI patients with an emphasis on the decreased of sense of loneliness and increased communicative initiatives.

Regarding studies that have used anthropomorphic robots, one study used KABOCHAN, a robot with childlike features, in which participants were free–after a training session–to interact with the robot in individual sessions [34]. Participants' caring behaviors toward the robot were observed. Qualitative analyses found caring behaviors toward the robot and good user acceptability of the robot. Another study that used the KABOCHAN again in individual sessions with MCI in dementia care residences found improved sleep, feeding and language production, and reduced anxiety [22]. Another study that employed the NADINE robot in individual sessions where participants could ask it questions and ask to watch videos or play music assessed psychological well-being and its effects on impaired cognitive and social

domains [23]. Results showed improvement in psychological well-being and some unspecified cognitive abilities. Another study explored in individual sessions in the apartments of an elderly residence during lockdown the effects of the JAMES robot [25] on feelings of loneliness. The results showed a reduction in feelings of loneliness and social isolation. Overall, these studies aimed to assess the acceptability of robots by MCI patients in care settings, also showing improvements on feelings of loneliness and general psychological well-being as secondary outcomes.

Regarding interventions that offered group activities, one study used NAO in multimodal activities (physical, cognitive and social) to decrease patients' feelings of apathy in pairwise activities in which the therapist was also present [42]. Results showed that participants felt more engaged (i.e., visual, verbal, and behavioral engagement) in activities where the NAO was activated compared with parts of sessions where only the therapist was present. One study used SOPHIE and JACK to improve the quality of life (i.e., reduced feelings of loneliness and psychological well-being) of residents of a dementia nursing home [40]. The robots offered musical quizzes and games (i.e., Bingo) in groups. Qualitative analyses of the sessions found increased involvement in the games proposed by the robots with a positive attitude and improved social skills defined as interactions among residents and between residents and therapists. Another study used MATILDA robot for games (e.g., Bingo), music quizzes, and orientation activities in groups, showing an increase in social involvement among individuals compared to the baseline condition in which the robot was not present [41]. One study using PEPPER involved MCI in small groups where caregivers and the therapist were present [37]. In these sessions, the robot played some of the participant's favorite songs and asked them to recognize the song and evoke memories related to this song. The results showed increased positive emotions in patient-caregiver interactions and elicited autobiographical memory. One study compared the effect that interactions with different robots (NAO and PARO) and a dog had on psychiatric symptoms [24]. Patients performed individual therapeutic activities (e.g., identifying numbers, words and colors using flash cards), showing an improvement in apathy in the group with NAO and a reduction in disturbing behaviors at night in the group with PARO. The use of the robots in the patient groups mainly indicated an increase in interactions between resident of the care homes and between patients and therapists. No direct results on the effect from the point of view of social cognition were considered as an effect of the interventions.

## 4. Discussion

MCI is a borderline clinical condition between healthy aging and the development of more severe neurodegenerative conditions, such as Alzheimer Disease. To date, only nonpharmacological interventions can slow down cognitive decline and support residual social skills. Current technological development enabled the possibility of implement classical paper-and-pencil interventions through therapist via anthropomorphic and zoomorphic social robots. Internationally, the recommendation to use social robots is related to the opportunity to maximize the positive effects of cognitive and social interventions for MCI because they could always be available to people and execute protocols systematically in different settings. However, today's scientific picture on this topic is not entirely clear, and it was therefore necessary to systematically analyze the studies conducted so far through social robots with MCI to provide suggestions for future studies from both a robotic and clinical point of view. The present systematic review focused on two domains that are particularly relevant to MCI functioning and are progressively impaired: cognitive and social domains. The studies identified for this review were mainly aimed at analyzing whether and how much training employing social robots could improve cognitive and social skills in MCI.

The discussion will be organized by indicating on the one hand the state of the art on the use of robots and the types of interactions implemented for MCI interventions and on the other hand identifying the clinical limitations.

## 4.1 Social robots in cognitive and social interventions

Regarding cognitive interventions, the robot was used only in individual sessions with multi-domain cognitive exercises, and this is in line with classic intervention protocols in which activities are proposed individually and, given the complexity of the clinical picture, focus on multiple cognitive dimensions. Conversely, in social interventions, robots were used in group sessions whose main goal was to solicit the interest of participants and be mediator/animator of social exchanges among them.

In general, in cognitive interventions the robot is used as a device to administer tasks, proposing activities that the participant needs to complete. These tasks by stimulating specific cognitive abilities enable their enhancement. Studies have mainly focused on two cognitive domains: memory and executive functions (i.e., working memory). With respect to memory, tasks concerned music quizzes (i.e., recognizing the title of a song and associating the song with the singer) and quizzes on selected texts (e.g., answering questions about them and remembering key words), while executive functions were not directly addressed by cognitive trainings but were considered indirectly in social trainings. Specifically, in social interventions, the game of bingo was implemented as it allows training working memory by remembering the numbers of the draws. With respect to social interventions, these were implemented with both anthropomorphic and zoomorphic robots, while cognitive interventions only anthropomorphic robots were adopted. Zoomorphic robots due to their animal features were not considered tools to support cognitive functions probably because if they spoke, they would lose their resemblance to the animal. This limitation from the cognitive side, on the other hand, represents a strength for social interventions because their characteristics solicit caretaking of the robot by participants supporting relational components. More specifically, zoomorphic robots are particularly relevant in more severe dementia conditions while patients with MCIs who have different residual abilities, the use of anthropomorphic robots may be equally effective. This bias could be due to advanced cognitive impairment that reduces initiative in interactions and, therefore, it is easier to have an interaction on the sensory level occurring with zoomorphic robots that can be picked up and stroked.

Although most of the studies claimed to examine the effects of interventions with social robots, they actually examined the acceptability of the robots by the participants. Although the studies presented patients with different levels of cognitive decline, in all studies there were no episodes of rejection toward social robots regardless of the type of robot used. Even in cases characterized by a more severe clinical picture, both anthropomorphic and zoomorphic robots were easily integrated into daily and rehabilitative activities and generated curiosity fostering greater involvement in the therapeutic sessions. Despite these very positive results related to the acceptability of robots by the elderly, it is not possible to generalize the results related to cognitive and social interventions.

## 4.2 Clinical issues

From a clinical perspective, the studies report several methodological problems, the main ones are: clinical heterogeneity of samples; absence of standardized, pre/post-intervention and brain assessment protocols through functional techniques; and absence of control samples.

With respect to clinical samples, studies are mostly presented with mixed patient groups (MCI and mild/moderate/severe dementia), and those with only patients with MCI do not

report specifics regarding the most impaired areas, except for one study with amnestic patients (aMCI). Prospectively, studies should build their samples by considering only MCI clinical populations by specifying the clinical subpopulation (i.e., amnestic, non-amnestic, and multi-domain). This would allow the creation of customized interventions that could have more precise effects based on the specific needs of different clinical subpopulations. In fact, the studies analyzed in this review indiscriminately used cognitive exercises on memory without considering that MCI patients might have memory skills that are still functioning, but instead need specific exercises, for example, on executive functions. The problems in detecting clinical subpopulations of MCI also relates to the heterogeneity of the assessments used and the absence of functional brain assessments. Most importantly, not all studies report the assessment procedures adopted, and those studies that have reported them do not present structured protocols integrated with functional brain techniques. Future studies should detail more precisely the assessment tools adopted and consider assessments at different levels, including neurofunctional ones. The absence of control groups is an important methodological limitation because it results in a reduction in the generalizability of the positive results of the cognitive and social interventions. Future studies should consider including homogeneous MCI samples as control groups or compare different clinical populations grouped homogeneously with respect to the characteristics of the MCI sample involved (e.g., mild dementias with specific memory issues).

### 4.3 Robotics issues

Regarding the robotic area, there are two critical aspects to highlight: the variety of robots used and the use of the Wizard of Oz technique for all interventions. In recent years, it has emerged clearly in the literature on human-robot interaction how the type of robot strongly affects the perception and interactions with humans [43,44]. A few studies have tried to address this issue by comparing different robots (i.e., different anthropomorphic robots or an anthropomorphic robot vs. a zoomorphic robot), but the results do not present clear results because specific analyses were not carried out and, therefore, it remains open whether some social robots may be more effective than others in performing cognitive and social interventions. Another issue is the use of Wizard-of-Oz technique in interactions, highlighting a reduction in the scalability of these interventions. As a matter of fact, it would be difficult in specialized care institutions and people's homes to use robots if the programs must necessarily be run by technicians.

### 5. Conclusion

This systematic literature review following the PRISMA guidelines examined cognitive and social interventions through social robots with MCI patients. The overall goal was to explore the state of the art and to identify suggestions for future research in both robotics and clinical fields. In general, the studies showed that both anthropomorphic and zoomorphic social robots are accepted by MCI patients. On the cognitive side, although the results showed that social robots can support some cognitive functions (e.g., memory and executive functions), the intervention protocols are not yet clinically standardized and mainly represent feasibility studies on the implementation of cognitive stimulation exercises with social robots. On the social side, studies have used robots mainly as mediators of the relationship between patients with MCI and facilitators of group sessions. However, there is no evidence of the effectiveness of robots on the basic social-cognitive skills (e.g., ToM) of patients with MCI, which the clinical literature has revealed to be susceptible to decay. With respect to the types of social robots used, we are still on a universalistic side, where we refer to robots in the singular without considering that different social robots have different impacts on how users perceived them. This

is even more important in care settings where different devices can affect the effectiveness of interventions. Furthermore, the preference for the Wizard-of-Oz technique in interventions demonstrates the need to develop social robots with semi-autonomous interactive sequences. This would have a positive effect on the scalability of interventions in settings where technicians might not be present (e.g., in the homes of patients with MCI).

On the clinical side, there is a need to promote more accurate patient assessment protocols so that the effects of interventions in different clinical subpopulations with MCI can be more accurately identified. In addition, it would be highly desirable to identify pre- and post-intervention assessment protocols that can precisely identify the effects of interventions on the most impaired cognitive and social domains. Standardized assessment protocols would also allow the identification of homogenous samples of MCI patients and their control groups, which would be essential for generalizing the results of interventions. Prospectively, socio-cognitive and neurofunctional assessments (e.g., MRI) should be considered in addition to cognitive assessments.

In conclusion, this literature review systematized the state of the art of Social Assistive Robotics in patients with MCI from both robotic and clinical perspectives, highlighting promising results of social robots in cognitive and social interventions while critically reporting insights for future research by recommending greater synergy between clinicians and robotic researchers.

## Supporting information

**S1 Fig. This figure shows the percentage of studies in terms of risk of bias (Low risk in green; Some concerns in yellow; High risk in red) subdivided by RoB 2 scale macro-category.**
(DOCX)

**S2 Fig. This figure illustrates the ratings of each macro category of RoB 2 and the final computation of risk of bias ("Overall").** The rating levels are divided into: Low risk (green); Some concern (yellow); High risk (red).
(DOCX)

**S3 Fig. This figure illustrates the ratings of each study subdivided for every item of the GRACE scale.** The rating levels are divided into: Yes + (green); Not applicable (yellow); No, or not enough information in article (red).
(DOCX)

## Author Contributions

**Conceptualization:** Giusi Figliano, Federico Manzi, Davide Massaro.

**Investigation:** Giusi Figliano.

**Methodology:** Giusi Figliano, Federico Manzi, Andrea Luna Tacci, Antonella Marchetti, Davide Massaro.

**Project administration:** Giusi Figliano, Federico Manzi, Davide Massaro.

**Supervision:** Giusi Figliano, Antonella Marchetti, Davide Massaro.

**Writing – original draft:** Giusi Figliano, Federico Manzi, Andrea Luna Tacci, Davide Massaro.

**Writing – review & editing:** Giusi Figliano, Federico Manzi, Andrea Luna Tacci, Antonella Marchetti, Davide Massaro.

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
