## [Decision Letter · Decision Letter 0]

24 Jul 2023

PONE-D-23-10921Ageing society and the challenge for social robotics: A systematic review of Socially Assistive Robotics for MCI patients.PLOS ONE

Dear Dr. Figliano,

Thank you for submitting your manuscript to PLOS ONE. After careful consideration, we feel that it has merit but does not fully meet PLOS ONE’s publication criteria as it currently stands. Therefore, we invite you to submit a revised version of the manuscript that addresses the points raised during the review process.

We look forward to receiving your revised manuscript.

Kind regards,

Mert Tanal

Academic Editor

PLOS ONE

Journal Requirements:

"No"

"No"

Additional Editor Comments :

As the reviewer mentioned, the abstract should be revised with improving the quality of language with minor revisions. Best regards,

Reviewers' comments:

Reviewer's Responses to Questions

**Comments to the Author**

1. Is the manuscript technically sound, and do the data support the conclusions?

Reviewer #1: Yes

2. Has the statistical analysis been performed appropriately and rigorously? 

Reviewer #1: Yes

3. Have the authors made all data underlying the findings in their manuscript fully available?

Reviewer #1: Yes

4. Is the manuscript presented in an intelligible fashion and written in standard English?

Reviewer #1: Yes

5. Review Comments to the Author

Reviewer #1: This is an interesting article showing the 'effects' of using a socially assisted robot in people with MCI; It provides an overview of the different methods that has been used while implementing a robot and shows also the shortcomings. The procedure for a systematic review has been followed rigorously. One importent issue that should be tackled is the abstract. In the abstract no information is given about the method used. That should be added.

6. PLOS authors have the option to publish the peer review history of their article (what does this mean?). If published, this will include your full peer review and any attached files.

Reviewer #1: **Yes: **Dominique Van de Velde

---

## [Author Response · Author response to Decision Letter 0]

3 Aug 2023

Dear Editor,

as requested by the reviewer, we have provided more details on the methods used in the abstract.

I has also added two references to line 77 and consequently updated the numeration of the bibliography.

The authors received no specific funding for this work.

---

## [Editor Report · Decision Letter 1]

11 Oct 2023

Ageing society and the challenge for social robotics: A systematic review of Socially Assistive Robotics for MCI patients.

PONE-D-23-10921R1

Dear Dr. Figliano,

We’re pleased to inform you that your manuscript has been judged scientifically suitable for publication and will be formally accepted for publication once it meets all outstanding technical requirements.

Kind regards,

Mert Tanal

Academic Editor

PLOS ONE

Additional Editor Comments (optional):

Well revised manuscript. Best regards
---

## [Editor Report · Acceptance letter]

22 Nov 2023

PONE-D-23-10921R1 

Ageing society and the challenge for social robotics: A systematic review of Socially Assistive Robotics for MCI patients. 

Dear Dr. Figliano:

I'm pleased to inform you that your manuscript has been deemed suitable for publication in PLOS ONE. Congratulations! Your manuscript is now with our production department. 

Kind regards, 

on behalf of

Dr. Mert Tanal 

Academic Editor

PLOS ONE